# Examining Facilitators of HPV Vaccination Uptake in Men Who Have Sex with Men: A Cross-Sectional Survey Design

**DOI:** 10.3390/ijerph17217713

**Published:** 2020-10-22

**Authors:** Selma Stearns, Samantha L. Quaife, Alice Forster

**Affiliations:** Research Department of Behavioural Science and Health, UCL, Gower Street, London WC1E 6BT, UK; selma.stearns.18@ucl.ac.uk (S.S.); samantha.quaife@ucl.ac.uk (S.L.Q.)

**Keywords:** men who have sex with men, human papillomavirus, sexually transmitted infections, sexual health, vaccination

## Abstract

Men who have sex with men (MSM) in England are eligible for vaccination against human papillomavirus (HPV) via specialist sexual health services and HIV clinics. Uptake among clinic attendees is incomplete, but the reason for this is unclear. We do not know who is accessing and being offered the vaccine. This cross-sectional study conducted in England examined socio-demographic correlates of vaccine uptake for MSM and how frequently the vaccine is being offered in clinics. MSM completed an online questionnaire asking about socio-demographic characteristics, whether they had recently attended a sexual health or HIV clinic, and if so, whether they had been offered the vaccine, and vaccination status. Around 52% of MSM (*N* = 115; mean age = 30.2) had received at least one dose of the vaccine, and 70% of clinic attendees had been offered the vaccine. MSM were more likely to have initiated the vaccine series if they were homosexual (versus bisexual; OR: 5.22; 95% CI: 1.55–17.51) or had heard about the vaccine from one or two types of sources (versus no sources: OR: 14.70; 95% CI: 4.00–54.00 and OR: 26.00; 5.74–117.77 respectively). Initiation was not associated with age, ethnicity, education level, or number of sexual partners. Hepatitis B vaccination status was associated with vaccination initiation only in unadjusted models. The majority of eligible MSM are being offered the vaccine in clinics. Socio-demographic differences in uptake of the HPV vaccine among MSM may lead to inequalities in HPV-related disease.

## 1. Introduction

Human papillomavirus (HPV) is a common sexually transmitted infection that can lead to cervical, penile, anal, and oral cancers and genital warts [1]. The National Health Service (NHS) in England currently offers the Gardasil quadrivalent vaccine, which protects from four types of HPV: 6, 11, 16, and 18. Types 16 and 18 are the cause of more than 70% of cervical cancers in the UK, types 6 and 11 cause around 90% of genital warts [2], and type 16 has been detected in about 70% of anal cancers [3]. Since 2008, girls aged 12–13 have been vaccinated for HPV in schools in England, and it had been presumed that boys are indirectly protected from HPV through this scheme through “herd protection” [1,4]. However, men who have sex with men (MSM) receive much less indirect protection from this programme than heterosexual men [4], and there is no catch-up programme for the new boys’ school-based vaccination programme [2], so boys over age 13, especially MSM, are still at risk. MSM also have an increased risk of HPV infection (one study detected HPV types 16 and/or 18 in 37% of MSM 16–30 years old [5]) and associated diseases; one meta-analysis found that anal HPV infection and anal cancer precursors were very common in MSM [6]. Rates of anal cancer are over 17 times higher among MSM relative to heterosexual men [7], and the burden of anal cancer is exceptionally high among HIV-positive MSM [8]. The NHS is newly implementing a national HPV vaccination programme for MSM in England using the quadrivalent vaccine. Note that the more protective Gardasil9® can be considered in the next round of tendering for the HPV programme, as it was not licenced in time for the current contract [9]. Public Health England (PHE) conducted a pilot study for the programme, in which less than half of eligible MSM (defined as those who have ever reported being MSM aged up to and including 45 years attending pilot site clinics on or after the date of first implementation at the clinic in question, and who did not initiate vaccination prior to the start of the pilot) received the vaccine [4]. There is no cost to these MSM to receive the vaccine. It is essential to understand which factors influence HPV vaccine uptake to make improvements through interventions or policy.

Research has recently been conducted in the United States examining factors associated with HPV vaccination uptake in MSM [10,11,12,13]. This research has revealed potentially relevant socio-demographic factors associated with vaccination, including white ethnicity, higher educational attainment, a higher number of sexual partners, and gay sexual identity. Having recently visited a healthcare professional, having disclosed same-sex attraction to a healthcare professional, and a recommendation from a healthcare professional have also been associated with vaccination update. Finally, socio-cognitive variables associated with vaccination include positive perceived norms about HPV and greater knowledge about HPV. Factors relevant to vaccination in this population may also come from health behaviour models: Theory of Planned Behaviour variables were associated with HPV vaccination in MSM [10], and Health Belief Model variables were associated with hepatitis B vaccination uptake in MSM [14]. However, differences between the United States and England, including in the healthcare systems, limit the applicability of these results. One recent study conducted a questionnaire and focus groups with young MSM in the UK to understand their knowledge and attitudes towards the vaccine [15] but did not statistically examine barriers and facilitators. Research specifically examining correlates of HPV vaccine uptake in MSM in England is required. Furthermore, previous research on uptake has mostly examined young MSM under age 26 [10,11,12,13,15], but the HPV vaccine is being offered in England to MSM up to age 45 [9]. In order to understand relevant factors in the entire group of eligible individuals, it is necessary to include older MSM.

There is evidence that MSM in England are willing to receive the vaccine [15], and the pilot suggests that few MSM refuse the vaccine when it is offered [4], yet uptake is not optimal. The present study extends previous research to a sample of men of varied ages in England, with the following research questions:How many eligible MSM are being offered the HPV vaccine?Which factors are associated with initiation of the vaccination schedule?

## 2. Materials and Methods 

Participants, recruited online using convenience sampling, completed a cross-sectional questionnaire assessing vaccination status and potentially related variables. The UCL Research Ethics Committee granted approval (project ID: 15043/001).

Participants (*n* = 115) were MSM in England who either received information from a Lesbian, Gay, Bisexual, and Transgender (LGBT)-associated group or used Grindr, a popular MSM dating application. Eligibility requirements were living in England, identifying as male, being attracted to males, and being age 18–45 (as the vaccine is offered to men up to age 45). 

Online data collection occurred from 8 April to 13 June 2019. Participants were offered a drawing for a £50 voucher as an incentive. In total, 172 LGBT and MSM groups in England were asked to advertise the study through email and social media (Twitter). Groups were contacted if they were near the cities included in the PHE pilot [4] to increase the probability that participants had attended clinics that offered the vaccine. In order to improve the representativeness of the sample, bisexual and Black, Asian, and minority ethnic (BAME)-specific groups were contacted to increase recruitment from those populations. Tweets regarding the research were also shared by individuals, including academics and healthcare professionals (HCPs). Participants were also recruited through an advertisement on the popular MSM dating application Grindr (3 May to 28 May 2019).

A researcher met with an individual from the target group to ensure the questionnaire was phrased comprehensibly and appropriately. The final questionnaire had 43 questions and was created to assess factors shown to be relevant in previous research (see Appendix A). Participants were asked about sexual health clinic attendance (“Have you attended a sexual health or HIV clinic since summer 2016?”) and HPV vaccine status. Only socio-demographic and some healthcare-related dependent variables are reported here: age (“What is your age?”), ethnicity (“What is your ethnic group?”), education (“Please select the highest level of education you have completed.”), sexual orientation (“Which of the following most closely matches your sexual orientation?”), number of sexual partners (“In your lifetime, with how many different people have you had vaginal, anal, or oral sex? This question is optional.”), number of information sources (“Have you ever heard about the HPV vaccine from any of these sources? Check all that apply.”), and Hepatitis B vaccination status (“The hepatitis B vaccine is a series of three injections. How many hepatitis B vaccine injections, if any, have you had?”). The item on information sources was recoded to reflect the number of information sources the participant had heard from about HPV for uni- and multivariable analyses.

The questionnaire was hosted on the site Opinio and included consent and information pages. Participation was anonymous to reduce response bias. Recruiting participants online can be beneficial as it allows the researcher to reach a large group of people without geographical restrictions. Participants may also feel more inclined to take part in online anonymous research rather than in person, especially when research concerns sensitive information such as sexual health. 

Participants were provided with detailed information and links at the end of the questionnaire, including about how to receive the vaccine [16,17], side effects [18], and possible risks involved [19].

Participants were not included if they did not complete consent, did not meet eligibility criteria, did not respond following eligibility assessment, or did not answer enough questions to determine vaccination status. If participants had answered enough questions to determine status, they were included in analyses with pairwise deletion. The analyses were as follows:
How many eligible MSM are being offered the HPV vaccine?: Of those who had attended a clinic, the proportion of those who were offered the vaccine was calculated.Which factors are associated with initiation of the vaccination schedule?: We used univariable binary logistic regression to investigate which variables were associated with having received the vaccine (defined as receipt of ≥1 dose of HPV vaccine; *n* = 60) or not having received it (*n* = 55). Associations were examined for seven variables: age, ethnicity, sexual orientation, number of sexual partners, information sources, and hepatitis B vaccination. All variables associated with initiation at *p* < 0.05 were entered into a multivariable logistic regression analysis using the “enter” method for selection. 

## 3. Results

In total, 167 participants initiated the questionnaire, and data from 115 participants were included in final analyses. Descriptive statistics are given in Table 1, stratified by vaccination status.

The average age of participants was approximately 30 years, and the sample was mostly white (84.5%; *n* = 93), highly educated (74.5%; *n* = 82), and homosexual (82.6%; *n* = 95). Approximately 4% of participants had had sex with zero people (*n* = 3), and approximately 40% had sex with 51 or greater people (*n* = 30). Half of the participants had heard about the vaccine from one source (53.9%; *n* = 62). The counts of hearing about the vaccine from each of the following sources was as follows: doctor or healthcare provider (*n* = 57); friend or family member (*n* = 23); brochure or poster (*n* = 12); commercial or advert (*n* = 3); other (*n* = 20). Participants could select more than one source. Half of the participants had completed the hepatitis B vaccine course (49.1%; *n* = 54), and this was highest among those who had had two doses of the HPV vaccine (76.5%; *n* = 13). Most participants had been to a clinic (80.4%; *n* = 90).

Frequencies of vaccination status were as follows: 40.9% (*n* = 47) had not been offered the vaccine; 7.0% (*n* = 8) had been offered the vaccine but not received it; 12.2% (*n* = 14) had one dose of the vaccine; 14.8% (*n* = 17) had two doses; and 25.2% (*n* = 29) had three doses.

### 3.1. How Many Eligible MSM Are Being Offered the HPV Vaccine?

In total, 90 participants (80.4%) reported having attended a clinic. All participants fulfiled eligibility for the vaccine, and therefore, all participants who attended a clinic should have been offered it; 63 (70.0% of those who attended a clinic) said they had been offered the vaccine. Of those who attended a clinic, 30 (33.3%) had not received the vaccine, 7 (7.8%) had received one dose of the vaccine, 17 (18.9%) had received two doses, and 29 (32.2%) had completed the vaccine series. Seven (7.8%) had initiated the vaccine series but were unsure how many doses they had had. Therefore, 66.7% of participants who attended a clinic initiated the vaccine series. Overall, 52.2% of participants had initiated the vaccine series.

### 3.2. Which Factors Are Associated with Initiation of the Vaccination Schedule?

Table 2 presents the results of the univariable and multivariable regressions examining which variables were associated with initiation of the vaccine schedule. Odds ratios are presented with 95% confidence intervals. 

Three variables were significantly associated with vaccination in unadjusted models at *p* < 0.05. Homosexual men were more likely to have initiated the vaccine series than bisexual men (OR: 5.22; 95% CI: 1.55–17.51). Compared with those who had not heard about the vaccine from any sources, those with one type of source (OR: 14.70; 95% CI: 4.00–54.00), and two or more types of sources (OR: 26.00; 5.74–117.77) were more likely to have initiated the vaccine. Those who had completed the hepatitis B vaccination series were more likely to have initiated the HPV vaccine (OR: 3.82; 1.52–9.61). 

The multivariable model including the three significant variables was a good fit to the data (Hosmer and Lemeshow test (χ2(6) = 5.42, *p* = 0.491)) and explained 40.2% of the variance. The effects of sexual orientation and number of information sources remained in the same direction and were statistically significant in the model. Hepatitis B vaccination status was no longer significant in the model.

## 4. Discussion

The present study was the first to statistically examine correlates of HPV vaccine uptake in MSM in England across a wide age range since the vaccination scheme was introduced.

Being homosexual increased odds of vaccination, which replicates some results from the USA [12]. This may indicate that homosexual men are more likely to be aware of the vaccine or request it from a healthcare provider, or potentially healthcare providers are more likely to offer the vaccine if a man discloses he is homosexual rather than bisexual. To improve uptake, campaigns and healthcare providers could encourage bisexual men to receive the vaccine. Further, qualitative work may be needed to understand acceptability and receptivity among MSM with diverse sexual orientations to understand how best to communicate the offer in clinics. A correlate of vaccine uptake was the number of different types of information sources from which a participant had heard about the vaccine. This may suggest that in addition to HCPs discussing the vaccine with patients, those who are exposed to vaccine information through friends, family, and public health campaigns are more likely to be vaccinated. Having received three doses of the hepatitis B vaccine increased odds of vaccination in univariable analysis, which replicates research from the United States [11]. 

Socio-demographic variables implicated in previous research, such as ethnicity (e.g., [10]), education (e.g., [12]), and number of sexual partners (e.g., [11]) were not significantly associated with uptake in the present research. Age was also not a significant correlate of uptake, which may indicate that results from research concerning young MSM may be applicable to older men as well. All participants in the present research were eligible for the vaccine, as they were all MSM under the age of 45. Around 41% of participants (*n* = 47) had not been offered the vaccine. Some participants had not attended clinics and therefore could not have been offered the vaccine; however, some participants had attended a clinic and not been offered it. A small recent study found that most young MSM want the vaccine and would like more information about it [15]; therefore, it is important to assist this group to move to the next stages by ensuring the vaccine is offered. A similar proportion (40%; *n* = 46) of participants had two or three doses of the vaccine. This indicates that there may be a large population of individuals who successfully receive the vaccine, and importantly, this group “maintains” the behaviour, as they return for the second and third doses, which are required for effectiveness.

The NHS expected an uptake of 60% for the first dose [9]; in the present research, approximately 70% of those who had attended a sexual health clinic had been offered the vaccine, and approximately 67% reported having initiated the vaccine. The NHS expected an uptake of 55% for the second dose and 50% for the third dose; in the present research, approximately 51% and 32% of those who attended a clinic had at least two doses and completed the series respectively. Therefore, initial uptake may exceed NHS expectations, while targets for maintenance of the behaviour may be optimistic. However, biases in the sample characteristics including ethnicity and a convenience sampling approach could be leading to an overestimation of uptake for the wider MSM population.

### Limitations

The current sample may not represent those who do not engage with LGBT groups, those from ethnic minority backgrounds, or those with fewer formal qualifications. We did not include people under 18 years old, and the vaccination scheme does not have a lower age limit. This means we do not have representative views of very young MSM. MSM often have their first sexual contact with a man before disclosing their sexual orientation to an HCP [20]; therefore, it is important that uptake is optimised for younger MSM. By using online advertisements for recruitment, there is the possibility of non-response bias if those who click on advertisements for participation differ significantly from those who did not [21]. This can mean that the sample is not representative of the population it seeks to examine, and this may be a particular issue in this population, as MSM who may be less open about their sexuality would not be reached. This design could have been improved if it were possible to obtain data from men who it is known have had sex with men, such as through medical disclosure, rather than requiring interaction with or participation in Grindr or online LGBT groups. The design would also have been stronger using representative sampling methods rather than convenience sampling and self-report measures.

There may be associations that this research was underpowered to detect due to the sample size. However, this research was exploratory, and while it may not provide concrete effect sizes, it implicates some important targets to increase vaccination uptake in MSM.

## 5. Conclusions

Approximately half of MSM in the present sample had initiated the vaccine series, suggesting that the NHS overall targets for uptake are not unreasonable. The present research highlights some potential targets for intervention, although future longitudinal examination may clarify the direction of effects.

## Figures and Tables

**Table 1 ijerph-17-07713-t001:** Descriptive statistics by vaccination status. *n* = 115 ^1^.

Variable	Had Not Been Offered Vaccine (*n* = 47)	Had Been Offered but Had Not Received (*n* = 8)	Had One Dose of Vaccine (*n* = 14)	Had Two Doses of Vaccine (*n* = 17)	Had Three Doses of Vaccine (*n* = 29)	Total 1 (*n* = 115)
Age, mean (SE)	30.13 (1.23)	32.50 (2.78)	31.57 (2.28)	34.24 (1.78)	26.66 (1.12)	30.20 (0.74)
Ethnicity, *n* (%)						
White	37 (84)	5 (71)	13 (93)	13 (76)	25 (89)	93 (85)
BAME	6 (14)	2 (29)	1 (7)	4 (24)	2 (7)	15 (14)
Other/prefer not to say	1 (2)	-	-	-	1 (4)	2 (2)
Education, *n* (%)						
Below university	11 (25)	2 (29)	6 (43)	5 (29)	4 (14)	28 (25)
University and above	33 (75)	5 (71)	8 (57)	12 (71)	24 (86)	82 (75)
Sexual orientation, *n* (%)						
Homosexual	35 (74)	6 (75)	10 (71)	16 (94)	28 (97)	95 (83)
Bisexual	12 (26)	2 (25)	4 (29)	1 (6)	1 (3)	20 (17)
Number of sexual partners, *n* (%)						
0 partners	1 (4)	1 (25)	-	-	1 (4)	3 (4)
1–12 partners	16 (57)	-	1 (10)	-	3 (13)	20 (26)
13–50 partners	5 (18)	1 (25)	3 (30)	5 (45)	9 (39)	23 (30)
51^†^ partners	6 (21)	2 (50)	6 (60)	6 (54)	10 (43)	30 (39)
Information sources, *n* (%)						
No sources	23 (49)	3 (38)	2 (14)	1 (6)	-	29 (25)
1 source	19 (40)	4 (50)	7 (50)	13 (76)	19 (66)	62 (54)
2^†^ sources	5 (11)	1 (13)	5 (36)	3 (18)	10 (34)	24 (21)
Hepatitis B vaccination, *n* (%)						
Did not initiate series	18 (41)	3 (43)	4 (29)	3 (18)	4 (14)	32 (29)
Initiated series	10 (23)	2 (29)	4 (29)	1 (6)	7 (25)	24 (22)
Completed series	16 (36)	2 (29)	6 (43)	13 (76)	17 (61)	54 (49)
Clinic attendance, *n* (%)						
Attended	22 (50)	8 (100)	14 (100)	16 (94)	28 (97)	90 (80)
Not attended	22 (50)	-	-	1 (6)	1 (3)	22 (20)

^1^ Totals do not sum due to missing data. Missing data for each variable are as follows: ethnicity (*n* = 5), education (*n* = 5), number of sexual partners (*n* = 39), hepatitis B vaccination (*n* = 5).

**Table 2 ijerph-17-07713-t002:** Univariable and multivariable correlates of vaccine uptake. *n* = 115 ^1^.

Continuous Variable	Mean: Received (SE)	Mean: Did Not Receive (SE)	Univariable OR	
Age	29.95 (0.99)	30.47 (1.13)	0.99 (0.95–1.04)	
Categorical variable		**No. received vaccine/total no. in category (%)**	**Univariable OR** **(95% CI)**	**Multivariable OR** **(95% CI)**
Ethnicity				
White		51/93 (54.8)	1.00 (Ref)	
BAME		7/15 (46.7)	0.72 (0.24–2.15)	
Other/prefer not to say		1/2 (50)	0.82 (0.05–13.57)	
Education				
Below university		15/28 (53.6)	1.00 (Ref.)	
University and above		44/82 (53.7)	1.00 (0.43–2.37)	
Sexual orientation				
Bisexual		6/20 (30.0)	1.00 (Ref.)	1.00 (Ref.)
Homosexual		54/95 (56.8)	3.07 (1.09–8.69) *	5.22 (1.55–17.51) *
Number of sexual partners				
0 partners		1/3 (33.3)	1.00 (Ref.)	
1–12 partners		4/20 (20.0)	0.50 (0.04–7.00)	
13–50 partners		17/23 (73.9)	5.67 (0.43–74.38)	
51+ partners		22/30 (73.3)	5.50 (0.44–69.26)	
Information sources				
No sources		3/29 (10.3)	1.00 (Ref.)	1.00 (Ref.)
One source		39/62 (62.9)	14.70 (4.00–54.00) *	13.78 (3.53–53.88) *
Two or more sources		18/24 (75.0)	26.00 (5.74–117.77) *	33.48 (6.36–176.19) *
Hepatitis B vaccination				
Did not initiate series		11/32 (34.4)	1.00 (Ref)	1.00 (Ref)
Initiated series		12/24 (50.0)	1.90 (0.65–5.64)	1.06 (0.29–3.84)
Completed series		36/54 (66.7)	3.82 (1.52–9.61) *	2.52 (0.84–7.54)

^1^ Totals do not sum due to missing data on some variables. * Significant at *p* < 0.05.

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
