# Peer review of "Examining Facilitators of HPV Vaccination Uptake in Men Who Have Sex with Men: A Cross-Sectional Survey Design"

_ijerph, 2020, doi:10.3390/ijerph17217713_

Round 1
Reviewer 1 Report
Dear Authors,
the manuscript submitted treats a very interesting and current topic. However, many aspects should be addressed before to consider the paper suitable for potential publication.
Please find below my recommendations:
Introduction could be improved i.e. report similar studies in other country
Material and methods: convenience sample could lead to selection bias. Please add motivation to this selection method and limitation of the study
Results
table 1 improve readability of the results.
lines 118 to 133 insert percentage or number when appropriate
better explain table 2 and uni/multivariate analysis
Reviewer 2 Report
The manuscript by Sterans et al. describes valuable information regarding the features of individuals in a health clinic to describe the vaccination status. The presented information surely will help in the development of public health strategies aimed to increase the vaccination uptake in this population. The manuscript is well written and organized. But some issues need address before considering this manuscript for publication.
Major comments
- Authors need to include a better description of the statistical analysis they used in the study.
- The authors indicate the examination of barriers in vaccine uptake, but this is not completely addressed in the results and the discussion section. Do the authors have more information regarding the potential barriers in vaccine uptake in the screened population? For instance, annual income, geographic area, etc. This will improve the results and certainly needs to be discussed.
Minor comments
- In the abstract it would be great to indicate the OR and the CI of the MSM to compare with the other groups, since this is only indicated for one group.
- Line 36, it would be great to indicate whether they refer to the bi or tetra-valent vaccine and to include the % of cancer associated to the HPV types included in the vaccine, since the G9 vaccine covers more than 95%.
Reviewer 3 Report
MINOR COMMENTS
- Line 35: the newest vaccine is 9-valent - this describes the quadrivalent. is the 90-valent not available in the UK?
- Line 37: i think this should just be named as what it is - herd immunity
- Line 44: what was the eligibility? This is important to understand. Was there a cost? Was the eligilibity very restricted? etc. Need to list criteria.
- Should provide brief rationale for age 18-45 - presumably based on the eligibility requirements for the vaccine, but we don't know what those are
- Line 74: define 'BME'
- The authors seem to be using univariate and univariable, and multivariate and multivariable interchangeably (Lines 92-93, 104-107), and these are different. Need to correct this.
- Line 112: separated or stratified?
- Line 116 - 30 years
- Line 118 - according to the table, this would be >50 people, not >51 people
- Line 119 - these aren't rates. these are absolute counts. Should actually include proportions as well e.g. (n = 4;10%)
- line 124-126 - fix the formatting of this section. Too many colons! e.g. ... as follows: 40.9% had not been offered the vaccine; 7.0% had been offered...etc
- Line 128 - again, we don't know what the eligibility criteria was
- Lines 152-157 - this paragraph is unclear. Needs to be rewritten. Groups with most participants? Maintain the behaviour? Unsure what is meant here.
MAJOR COMMENTS
- Since this paper is presumably meant for an international audience, some brief background in the Introduction on current availability for HPV vaccine would be helpful - e.g. school-based programs for girls/boys since 20XX. There's mention of the boys school-based program and forthcoming MSM program, but it's unclear what currently exists. This is important because there is such a discrepancy in actual vaccine availability, national/other vaccine guidelines and recommendations, and vaccine indications as per manufacturer. It would provide important context.
- Line 41 - it is indeed true that MSM have much higher rates of anal HPV infection and pre-cancers. There's lots of data on this so this should be quantified to really show the burden e.g. 60-70% in HIV neg, and up to 80+% in HIV+. Also, though these things are important, the real issue is anal cancer, and we know that HIV+ MSM have rates of anal cancer 50-100x that of the general population. In terms of setting up the rationale of the paper, this would strengthen this significantly.
- Paragraph starting at Line 47: this is a brief review of the literature on barriers/facilitators to vaccine uptake, but NOTHING is mentioned. Yes, there is very little data here, but it would be helpful to mention some of the findings that have been noted e.g. health-care provider recommendation has been associated with vaccine uptake, cost has been a barrier. If there isn't enough data from studies on MSM, then it's reasonable in setting this paper up to include data from other studies of non-MSM.
- Table 1 - the column headings formatting is off
- Table 1 - need to define all acronyms e.g. PAPM, BAME, and need to be consistent with use of these (previously used BME).
- Table 2 - difficult to read. Again, the column headings are confusing - univariate p-value above the multivariate OR??
- Table 2 - suggest double-checking the math of the last item (i.e. completed series HBV Vaccine). It correctly states in the text that this was not significant (as the CI cross 1) but the p value is very low. There seems to be a discrepancy between the CI and the p-value. Double check this.
- DISCUSSION - start out with your most interesting finding and discuss this first. Then get into the other stuff.
- This merits mentioning again - the entire discussion could be framed and contextualized much better if the vaccine eligibility criteria were known.
- Line 170 - need to provide a bit more thought around why this is an important finding and what - if anything - it means. Both homosexual and bisexual men are MSM, so what do we need to do differently in approaching the vaccine conversation with these two groups
Round 2
Reviewer 1 Report
Dear Authors the carried changes have increased readable and impactful of your study. MSM today are still a category on the margins of society, with difficulties in accessing treatment, especially when a positive HIV status is associated. Studies evaluating their access to health services and their immunization status (as a risk category) are important and should be implemented in every country.
I congratulate you on your study. Sincerely.